

# Differences in unilateral chest press muscle activation and kinematics on a stable versus unstable surface while holding one versus two dumbbells

Jeffrey M. Patterson, Andrew D. Vigotsky, Nicole E. Oppenheimer and Erin H. Feser

Kinesiology Program, Arizona State University, Phoenix, AZ, United States

## ABSTRACT

Training the bench press exercise on a traditional flat bench does not induce a level of instability as seen in sport movements and activities of daily living. Twenty participants were recruited to test two forms of instability: using one dumbbell rather than two and lifting on the COR bench compared to a flat bench. Electromyography (EMG) amplitudes of the pectoralis major, middle trapezius, external oblique, and internal oblique were recorded and compared. Differences in range of motion (ROM) were evaluated by measuring an angular representation of the shoulder complex. Four separate conditions of unilateral bench press were tested while lifting on a: flat bench with one dumbbell, flat bench with two dumbbells, COR Bench with one dumbbell, and COR Bench with two dumbbells. The results imply that there are no differences in EMG amplitude or ROM between the COR bench and traditional bench. However, greater ROM was found to be utilized in the single dumbbell condition, both in the COR bench and the flat bench.

## INTRODUCTION

Both sport and various activities of daily living (ADLs) require individuals to create pushing forces while standing. Most noticeably, this is seen in highly physical sports, like football or wrestling, but the same relative movement is seen when someone pushes a car or moves furniture around their home (*Santana, Vera-Garcia & McGill, 2007*). Traditionally, in order to train the muscles involved in these movements, people perform exercises like the bench press. There are, however, inherent problems in using the supine bench press when trying to improve one's ability to create pushing forces when standing.

First, in a supine bench press, either a bar or two free weights being used simultaneously create the resistance in the exercise, making it a bilateral movement. Unfortunately, in most sports and ADLs, movements are predominantly, or entirely, unilateral (*Behm et al., 2005*). Thus, the principle of specificity is not being fully utilized and the effects of training may not fully crossover.

Submitted **31 August 2015**
Accepted **9 October 2015**
Published **27 October 2015**

Corresponding author
Andrew D. Vigotsky,
avigotsky@gmail.com

Secondly, there are many factors that limit one's ability to develop pressing strength while standing, including stability and neuromuscular control of the core musculature, the individual's weight, the base of support the individual utilizes, and the direction in which the pressing motion is performed (*Santana, Vera-Garcia & McGill, 2007*). Since a traditional bench press is not performed standing up, these factors are not addressed. In particular, any given individual can usually only produce a standing pushing force of about 40.8% of their body weight (*Santana, Vera-Garcia & McGill, 2007*), whereas it is not uncommon for an individual to be able to lift greater than their body weight in a supine bench press.

Additionally, when performing a bilateral, seated pressing motion, the EMG amplitude of core stabilizers, such as the rectus abdominis and external obliques, is reduced significantly when compared to a standing, unilateral pressing motion (*Saeterbakken & Fimland, 2012*). Since most ADLs and sport actions are performed unilaterally and while standing, training in this method would enhance specificity while also increasing activation and strengthening of the trunk stabilizers (*Behm et al., 2005*). It can be clearly seen that a supine bench press will not produce the most optimal results for enhanced pressing strength while standing in sport or ADLs.

One final drawback to the bench press is the limitation of a full range of motion (ROM). It is possible that the flat surface of the bench, on which a bench press is completed, may limit an individual's ROM depending on their shoulder width. If individuals are only completing the bench press movement to parallel, they are likely not training through their full ROM. Previous research indicates that training should occur through a person's entire functional ROM (*Brown & Vives, 2000*; *Massey et al., 2005*; *Pinto et al., 2012*).

Because traditional exercises like the bench press do not translate well into standing pressing strength, it is the goal of many researchers today to find ways to make the bench press more similar to an actual movement that would be performed outside of the gym. Researchers have found that exercising on unstable surfaces can increase electromyographic (EMG) amplitude of the core musculature, such as the rectus abdominis (*Van Dieen, Kingma & Van der Bug, 2003*), external obliques (*Kohler, Flanagan & Whiting, 2010*), internal obliques (*O'Sullivan, 2005*), and muscles along the spine; e.g., erector spinae and multifidus (*Kohler, Flanagan & Whiting, 2010*; *O'Sullivan, 2005*). Furthermore, it has also been proposed that instead of performing bench press exercises with two dumbbells simultaneously—even if alternating which arm presses as in a unilateral bench press—people should use only one dumbbell at a time (*Behm et al., 2010*). This is due to that fact that when performing a purely unilateral exercise, the weight creates a torque on the body that the core musculature must counteract, whereas if there is a dumbbell in the offhand it will already counteract that torque with its own torque, thus reducing the work the core must perform.

To create an unstable environment, some researchers simply use equipment designed to introduce instability to an exercise. One such piece of equipment is the CÖR$^{TM}$ Bench (COR Bench). The COR Bench is an inflatable 400-denier nylon urethane tube that is supported with a stand much like a normal bench. It is the combination of the rounded

surface of the COR Bench and the fact that it is filled with air (i.e., not solid) that create an unstable environment for the individual, which may, in turn, increase both ROM and activation; however, this has yet to be substantiated. Being able to work through a larger ROM and targeting more muscles for stabilization can provide greater neurological and neuromuscular stimulation (*Marinković, 2011*).

Therefore, the purpose of this study was to examine the following questions: does performing a chest press on the COR bench elicit greater EMG amplitudes of the core muscles, and possibly the prime movers, when compared to a traditional bench? Will a greater ROM be used on the COR bench than the traditional flat bench? And finally, will a greater ROM be used with one arm when compared with two?

## METHODS

### Experimental approach to the problem

To test whether or not the COR Bench provides a greater stimulus for resistance training, participants performed a one dumbbell and two dumbbell unilateral bench press on both the COR Bench and a traditional bench. Participants were asked to come to the lab to complete two different sessions, the first was to identify the participant's bench press one repetition maximum (1RM) on both the COR Bench and flat bench, and the second was to test muscle activation and the kinematics of their movement during unilateral bench presses with one and two dumbbells on the COR and flat benches.

### Subjects

Fifteen males (age, $M = 23.1$, range $= 20$–$28$ years; mass, $88.7 \pm 11.9$ kg; height, $183.8 \pm 5.6$ cm) and 7 females (age, $M = 20.9$, range $= 19$–$23$ years; mass, $56.8 \pm 5.9$ kg; height, $167.8 \pm 4.7$ cm) were recruited to participate over the course of a fall semester; however, the experiment was conducted in a laboratory (controlled environment) with a temperature of $\sim$23 °C. Participants were required to be 18–30 years old, have previous experience in participating in resistance training, be free from current or previous injury that may be aggravated by participating in upper extremity resistance training, and also be familiar with the bench press. All participants were recreationally active. Participants were asked to abstain from training for 24 h prior to testing. Before beginning, all participants completed an Informed Consent and Physical Activity Readiness-Questionnaire. The Institutional Review Board at Arizona State University approved this study and its relevant documents (IRB ID: 1211008510).

### Procedures

On the first day of testing, the participants' 1RM was predicted using a method proposed by *Mayhew et al. (1992)*. This method was chosen over others because of its accuracy, as determined by a meta-analysis done by *LeSuer et al. (1997)*. Participants' 1RM testing order was randomized. Following, each participant's 1RM could be mathematically predicted; male's 1RM for the flat bench was $65.93 \pm 17.66$ kg and for the COR Bench it was $57.38 \pm 17.40$ kg, and female's 1RM for the flat bench was $18.55 \pm 4.68$ kg and for the COR Bench it was $17.78 \pm 4.47$ kg. All loads are presented as the sum of both dumbbells (total,

combined load). Because certain loads could not be created with the dumbbells used in the study, actual weights used were slightly different than the mathematically predicted values. The actual weights used due to dumbbell limitations were as follows: male's 1RM for the flat bench was $65.72 \pm 17.77$ kg and for the COR Bench it was $58.78 \pm 18.76$ kg, and female's 1RM for the flat bench was $20.52 \pm 4.88$ kg and for the COR Bench it was $19.44 \pm 4.10$ kg.

The second day of testing consisted of the kinematic and EMG analysis. Disposable, self-adhesive, Ag/AgCl pre-gelled, bipolar electrodes (Noraxon Product #272; Noraxon USA Inc., Scottsdale, AZ), with a diameter of 1 cm and an inter-electrode distance of 2 cm, were placed on one prime mover muscle (right pectoralis major), one shoulder girdle stabilizer (right middle trapezius), and two muscles associated with core stabilization (left internal and right external obliques) (*Konrad, 2005*). The left internal and right external obliques were selected because they perform ipsilateral and contralateral rotation of the trunk, respectively. Given that the one dumbbell unilateral bench press trials were performed with the dumbbell in the right hand, these muscles should activate to counteract the tendency of the trunk to rotate to the right side.

Before electrode placement, the skin over each muscle was prepared for EMG analysis by removing any hair and abrading and cleaning the skin with alcohol wipes. The EMG signals, collected at 1500 Hz, were analyzed using MyoResearch 3.0 (Noraxon, USA 2012). The collected data was rectified, filtered using a 10–500 Hz bandpass filter, and smoothed using root mean square with a window of 50 ms. Average amplitude was then determined for the duration of each trial.

To assess the kinematics of the participant's movements, reflective markers were placed on the skin over the acromion process of the scapula and the lateral epicondyle of the humerus, allowing for the analysis of ROM during the movement. As the electrodes were reaching a stable impedance condition, the participants engaged in a brief warm-up consisting of 10 arm circles forward, 10 arm circles backward, and 10 wall push-ups. A camera was placed behind the bench to capture the motion of the participant's shoulder complex and their ROM. The shoulder complex ROM (SC ROM) was measured by subtracting the minimum from the maximum 3- point angle and averaging the value for the five trials. Each 3- point angle was determined from points at a line held vertical, the acromioclavicular joint, and the lateral epicondyle. This was thought to be the best angular representation of the shoulder complex as a whole (Fig. 1).

In a randomized fashion, each participant completed five repetitions at 70% of their 1RM of unilateral bench presses holding two dumbbells on both the COR Bench and the flat bench as well as five repetitions at 70% of their 1RM of unilateral bench presses holding one dumbbell on both the COR Bench and flat bench, for a total of 4 separate trials. When they were holding two dumbbells, they completed five repetitions for each arm in an alternating style, starting with both hands raised. In other words, starting with both hands in the air, their right arm came down first, then pushed up, followed by the left arm coming down, then being pushed back up, equaling one repetition per arm. All

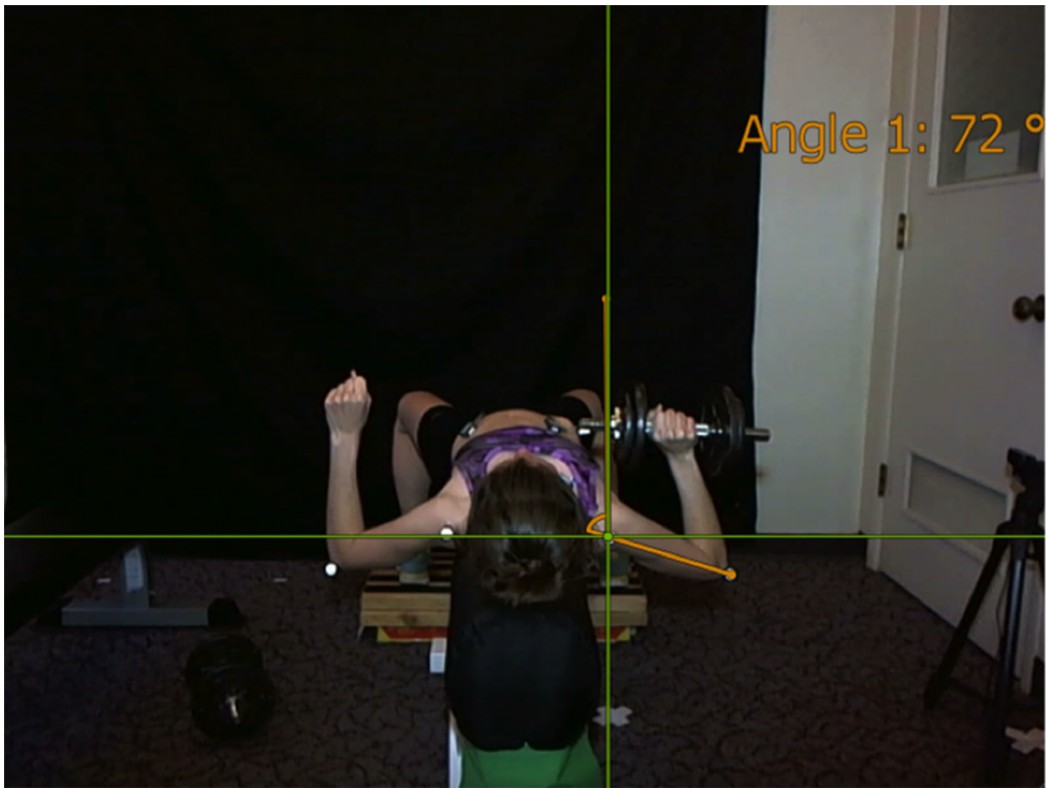

**Figure 1 Angular representation of the measurement of shoulder complex range of motion.** Differences in ROM evaluated by measuring angular representation of the shoulder complex: a 3-point angle determined from points at a line held vertical, the acromioclavicular joint, and the lateral epicondyle.

movements were controlled for speed with a 40 beats per minute tempo metronome (2 beats down, 2 beats up), and participants were allowed two minutes rest between sets.

There existed a difference in height between the two benches, so to control for any confounding factors, an elevated footrest was put at the end of the COR Bench to make the heights equal. Additionally, participants were required to keep their feet and legs in similar positions between both 1RM tests and all four experimental trials. This was accomplished by measuring the knee angle (via goniometer) and distance between the participants' feet (via tape measure) while lying on the flat bench, and both of these were hold constant throughout the entire experiment for each participant individually.

## Statistical analysis

A within-subject experimental design assessing EMG amplitude and movement kinematics was used to test the differences between the COR Bench versus flat bench and one dumbbell versus two dumbbell unilateral bench press. First, normality of the data was checked using the Shapiro–Wilk test. On parametric data, sphericity assumptions were checked using Mauchly's test. Parametric data were analyzed using a repeated-measures analysis of variance (ANOVA). If data were parametric but did not meet sphericity assumptions, Greenhouse-Geisser corrections to degrees of freedom were applied to the

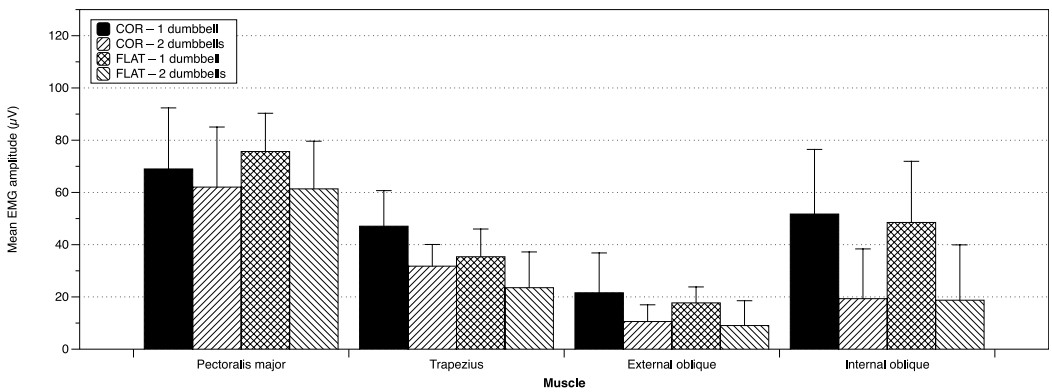

**Figure 2** **Mean EMG amplitude (±SD) in all 4 trials across all participants.**

ANOVA. Friedman's test was used to compare within-subject effects across groups for nonparametric data. For parametric data that required pairwise comparisons, two-tailed paired-samples $t$-tests were performed. Non-parametric data that required comparisons were compared using Wilcoxon paired-samples signed-rank tests. A Benjamini–Hochberg procedure was carried out to control for false discovery rate for all pairwise comparisons (*Benjamini & Hochberg, 1995*; *Colquhoun, 2014*). Alpha was set to $p \leq 0.05$, and $q$-value, or false discovery rate, was also set to 5%. Parametric effect sizes were calculated by Cohen's $d$ using the formula $d = \frac{M_d}{S_d}$, where $M_d$ is mean difference and $s_d$ is the standard deviation of differences (*Becker, 1988*; *Morris, 2007*; *Smith & Beretvas, 2009*). This method is slightly different than the traditional method of calculating Cohen's $d$, as it calculates the within-subject effect-size rather than group or between-subject effect sizes. Cohen's $d$ was defined as small, medium, and large for 0.20, 0.50, and 0.80, respectively (*Cohen, 1988*). Non-parametric effect-sizes were reported in terms of Pearson's $r$ ($r = \frac{z}{\sqrt{n}}$). Pearson's $r$ was defined as small, medium, and large for 0.10, 0.30, and 0.50, respectively (*Cohen, 1988*). In order to better represent within-subject changes, standard deviations (SD) were calculated based upon normalized data before graphing (*Loftus & Masson, 1994*); although, it should be noted that the within-subject normalization technique does not necessarily represent the pairwise comparisons made, as all four trials were taken into account.

## RESULTS

One participant's flat bilateral trial was corrupted (both EMG and ROM) due to a disk writing error, and one participant's middle trapezius EMG signal during the bilateral COR Bench trial appeared to be invalid, as there were impossibly high amplitudes, potentially due to pressure on the electrode, so these values were not included in the statistical analyses.

Using the calculated 1RM values, there was a $-11.59\%$ change in 1RM value when going from the flat bench to the COR Bench.

The mean EMG (±SD) amplitudes for each muscle under each testing condition are displayed in Fig. 2. The results of Friedman's test indicate that the differences observed between the four testing conditions in the middle trapezius [$\chi^2(3) = 16.200; p = 0.001$],

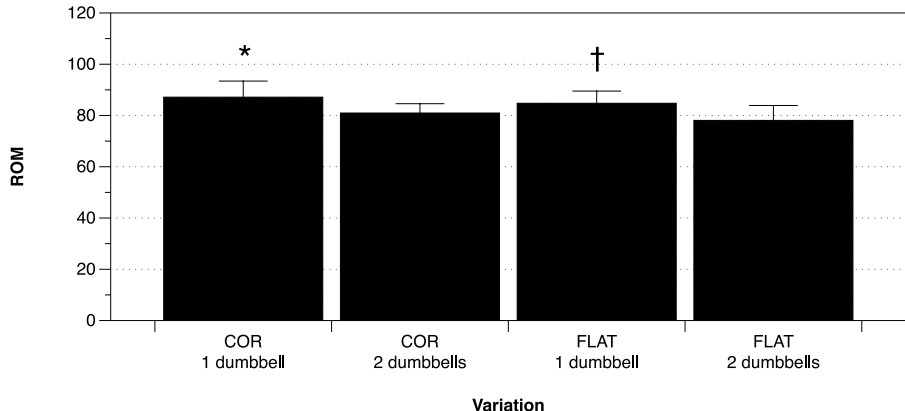

**Figure 3 Mean (±SD) shoulder complex ROM in all 4 trials across all participants.** (∗) denotes statistically greater shoulder complex ROM than using two dumbbells on the COR bench. (†) denotes statistically greater shoulder complex ROM than using two dumbbells on the flat bench.

external oblique [$\chi^2(3) = 38.556$; $p < 0.001$], and internal oblique [$\chi^2(3) = 39.758$; $p < 0.001$] were not due to chance alone. However, it cannot be said that the pectoralis major EMG amplitudes between all four testing conditions were statistically different [$\chi^2(3) = 7.444$; $p = 0.059$]. The results of the pairwise comparisons were used to address the following questions.

### Is there a difference in EMG amplitude when using one dumbbell on the COR Bench versus one dumbbell on the flat bench?

It cannot be concluded that any differences in EMG amplitude in the muscles analyzed [pectoralis major ($z = 0.859$; $p = 0.5207$; Pearson's $r = 0.19$), middle trapezius ($t(19) = 2.7428$; $p = 0.0516$; Cohen's $d = 0.61$), external oblique ($z = 1.157$; $p = 0.4236$; Pearson's $r = 0.26$), or internal oblique ($t(19) = 0.6912$; $p = 0.5431$; Cohen's $d = 0.15$)] during the one dumbbell trials on the COR bench versus the one dumbbell on the flat bench were not due to chance alone.

### Is there a difference in EMG amplitude when using two dumbbells on the COR Bench versus two dumbbells on the flat bench?

It cannot be concluded that any differences in EMG amplitude in the muscles analyzed [pectoralis major ($z = 0.684$; $p = 0.5431$; Pearson's $r = 0.16$), middle trapezius ($z = 1.807$; $p = 0.1600$; Pearson's $r = 0.43$), external oblique ($z = 1.751$; $p = 0.1600$; Pearson's $r = 0.40$), or internal oblique ($z = 1.046$; $p = 0.4431$; Pearson's $r = 0.24$)] during the two dumbbell trials on the COR Bench versus the two dumbbells on the flat bench were not due to chance alone.

### Is there a difference in SC ROM between flat vs. COR Bench?

The mean ± SD ROM for each testing condition is displayed in Fig. 3. The results of the ANOVA indicate that the four testing conditions were statistically different for shoulder complex ROM [$F(3, 54) = 10.256$, $p < 0.001$, $\eta^2 = 0.088$]. Pairwise comparisons revealed that a statistically greater ROM was utilized in one dumbbell trials when

compared to two dumbbells for both the COR bench ($t(19) = 3.826$; $p = 0.0066$; Cohen's $d = 0.86 (0.39, 1.32)$) and flat bench ($t(18) = 4.115$; $p = 0.0066$; Cohen's $d = 0.94 (0.46, 1.43)$). However, no statistical differences were observed between the COR bench and flat bench with two dumbbells ($t(18) = 2.509$; $p = 0.0657$; Cohen's $d = 0.58$) or one dumbbell ($z = 0.541$; $p = 0.5882$; Pearson's $r = 0.12$).

## DISCUSSION

It was hypothesized that the introduction of instability to a unilateral bench press movement would increase the EMG amplitude of the muscles being analyzed. In other words, an increase in co-contraction should be observed in the conditions with a less stable environment (COR bench) when compared to the conditions with a more stable environment (traditional bench). Additionally, it was hypothesized that an increase in ROM should be observed in the conditions performed on the COR bench due to its curved, narrow, and inflated surface. And lastly, it was hypothesized that participants would utilize a larger ROM with one arm than with two.

When performing the 1RM estimation test (*Mayhew et al., 1992*) on the flat and COR benches, there existed a $-11.59\%$ change in 1RM from the flat bench to COR Bench. This indicates that it would be considerably more difficult to lift a prescribed flat bench workload on the COR Bench. This is important for anyone who plans to use, or train a client/patient on, the COR Bench, as it shows that the instability of the COR Bench is substantial enough to warrant a reduction in the weight being chosen by a factor of about 10%. This is not uncommon, as performing exercises on an unstable surface can reduce the force output of the prime movers (*Behm & Colado, 2012*; *Hubbard, 2010*; *Marinković, 2011*; *Pontillo et al., 2007*).

The COR Bench was designed to provide instability to an exercise by not fully supporting the trunk like a traditionally used flat bench. Additionally, its design may allow for a greater ROM. Theoretically, these factors should elicit greater EMG amplitude of the core and shoulder girdle stabilizers and when performing lifts on the COR Bench when compared to a flat bench. However, in this present study, no statistical differences in EMG amplitude were observed for any muscle (pectoralis major, middle trapezius, external oblique, or internal oblique) when comparing the COR bench to its flat bench counterpart, for both one and two-dumbbell variations. These findings indicate that the COR Bench alone does not add enough instability to truly increase EMG amplitude in either core stabilizer muscles or prime movers analyzed in this study. This is similar to the results of *Goodman et al. (2008)*, who found that there were no differences in EMG amplitude across all muscles being analyzed when performing a chest press on a stable bench and an exercise ball, even when controlling for workload, in addition to *Uribe et al. (2010)*, who found no difference between dumbbell chest press on a bench and exercise ball with 80% 1RM. Again, however, while it cannot be said that the results were statistically different, there still existed greater EMG amplitude in the COR Bench trials compared to the flat bench trials with medium to large effect sizes. This may be an indicator that future research should be done to further determine the effectiveness of such equipment.

With regards to ROM, it does not appear that the narrower, air compressed surface of the COR bench transfers to greater ROM when compared to that of the traditional bench, for both one- and two-dumbbell variations. However, with both the COR bench and traditional bench, single dumbbell variations utilized a statistically greater ROM than did the two dumbbell variations. Such results suggest that the single dumbbell variation can be utilized to train with greater ROM, but whether these small ($\sim$7°), but statistically different, differences in ROM are sufficient to elicit superior training adaptations cannot be said for certain.

There are a few limitations that should be noted when interpreting the results of this present study. Firstly, no specific inclusion or exclusion criteria existed with regards to training age and frequency. It is possible that different responses may have been elicited from those with different training experiences, but this cannot be said for certain. Second, although participants abstained from training for 24 h prior to the second day of this study, some did complete the testing protocol on the preceding day. Therefore, it is possible that some participants may have been sore or fatigued on the EMG testing day, but because a within-subject design was utilized, participants were familiar with the movement, and the testing protocol was relatively light, it is unlikely that this would have had a substantial impact on the observed outcomes. In addition, the medium to large effect sizes observed for comparisons that did not reach the *a priori* alpha are indicative that this study may have been underpowered and type-II errors may be present, especially following the correction for multiple comparisons. Lastly, the 1RM estimation protocol utilized in this study has not been validated for the dumbbell bench press, and it is therefore possible that the estimated 1RM differed from participants' actual 1RM. Notwithstanding these limitations, our data are congruent with previous studies (*Goodman et al., 2008*; *Uribe et al., 2010*).

## CONCLUSION

In conclusion, the present study showed that the COR Bench did not prove to be an important of a source of instability, as it failed to elicit both greater EMG amplitude and ROM. Furthermore, it appears that using a single dumbbell may be a better option if larger ROM is desired. More comparative research—including training studies—involving athletes and rehabilitation patients with and without a history of upper extremity injury is warranted. Such further assessments will promote a better understanding of the other potential implications and advantages of the COR bench.

### Funding
The authors received no funding for this work.

### Competing Interests
The authors declare there are no competing interests.

## Author Contributions

- Jeffrey M. Patterson and Nicole E. Oppenheimer conceived and designed the experiments, performed the experiments, wrote the paper, reviewed drafts of the paper.
- Andrew D. Vigotsky analyzed the data, contributed reagents/materials/analysis tools, wrote the paper, prepared figures and/or tables, reviewed drafts of the paper.
- Erin H. Feser conceived and designed the experiments, performed the experiments, reviewed drafts of the paper.

## Human Ethics

The following information was supplied relating to ethical approvals (i.e., approving body and any reference numbers):

Arizona State University Institutional Review Board ID: 1211008510.

## Supplemental Information

Supplemental information for this article can be found online at http://dx.doi.org/10.7717/peerj.1365#supplemental-information.

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
