# Peer review of "Differences in unilateral chest press muscle activation and kinematics on a stable versus unstable surface while holding one versus two dumbbells"

_PeerJ, doi:10.7717/peerj.1365_

## Round 0.1 · original submission · Major Revisions

· Academic Editor

Major Revisions

Two expert reviewers have now evaluated your manuscript. Please see their comments below.

Reviewer 1 ·

Basic reporting

Line 62: Although 'traditional technique' may suggest that an individual stop their RoM when their humerus reaches parallel to the ground, there is plenty of other literature that suggests individuals to complete the full RoM. Perhaps this sentence should be removed.

Line 88-90: Previous work (See Behm JSCR 2006 review) suggests that individuals can exert greater forces in stable conditions which will result in the greatest gains in strength. Therefore, the authors should reconsider the wording of this sentence.

Experimental design

Predictive vs actual 1RM tests: All prediction equations in the cited literature are based on stable environments. Therefore, it is likely that the prediction equations are not entirely accurate for the unstable conditions. Since there is not a high risk in performing a 1RM test, the authors are requested to perform the experiments using actual 1RM values rather than predictive 1RM values. Perhaps this would rid any discrepancies that could arise from relative differences in intensities of weight used.

Validity of the findings

RoM: Were there significant differences in ROM between 1 and 2 arms? Is it possible that these differences in ROM can account for an increase in activation rather than instability?

Definitions of instability: Perhaps there could be more emphasis on the differences in the type of instability between the base of support (COR bench vs stable bench) and implement (1 vs 2 DBs). The authors are requested in substantiate differences in the type of instability these would create in the discussion.

Comments for the author

It appears that this MS was not thoroughly reviewed for sentence structure, grammar and flow prior to submission. Perhaps the MS could benefit from a thorough review from all authors.

Specific points:
Abstract:

"an increase in ROM was found between the stable and unstable conditions" - since the authors elude to two types instability (1 vs 2 bumbles and different benches), it is unclear if this sentence refers to both types of instability or just one, and if one, which one.

"the COR Bench does appear to be effective at increasing ROM due to bench
surface design differences" - Is this sentence supported in the MS? Perhaps, it would be better to remove this sentence if not.

Introduction:

Line 92: replace "is" with "was"?

Line 94: Perhaps the authors should avoid using the phrase "neurological stimulus to the core muscles" since they have no measure of the stimulus that is being induced to the core muscles. It is possible that this is a mechanical stimulus (i.e. perturbations due to instability)

Methods:

Were participants tested on back to back days? If so, how much time was between testing sessions?

Line 127: "1RM was determined" - perhaps this sentence is a misguiding description. The 1RM was predicted, not determined.

Line 131-132: Are these weights per dumbbell or combined of both arms? It may be beneficial to make that clear in text.

Line 193: Insert "were" between "assumptions" and "checked"?

Results:

Currently, the corrupted and invalid trials seem suspicious. Is there some way of including more information about the reasoning for the exclusion of these trials?

Discussion:

The phrase "due to chance alone" is used excessively throughout the MS. Perhaps, the authors could limit this phrase and use a smaller phrase such as "significant" when describing results.

·

Basic reporting

Overall the article was well very written and I believe it conforms to PeerJ's standards. My only comment regarding this section would be the legend on Figure 2 (EMG). On my first read through, I took the labels (ex. COR1/COR2) as meaning a numerical representation of the different experimental trials, and that the 1 and 2 would be defined in the figure caption. It was not immediately clear to me that "1" corresponded to unilateral pressing with one dumbbell and "2" to unilateral pressing with two dumbbells. Certainly not a major concern but I might suggest clarifying, either in the legend itself or in the figure caption, which bars correspond to which experimental trials.

Also, on line 171, I believe it should say, “both the COR bench 'and the flat bench' as well…”

Experimental design

The explanations of the experimental design were clearly written and easily understood. I applaud the authors for the detail of their investigation, which appears to have been rigorously conducted. For the most part, experimental factors were well controlled for. However, I have a few concerns, one major, regarding the protocols of this study.

1) My first concern is in regards to the trials of unilateral pressing with two dumbbells on both the COR and flat bench. In these two trials (COR with two dumbbells, flat with 2 dumbbells), participants were required to complete five repetitions with each arm in an alternating style. As I understand it, this differed from the remaining two trials of unilateral pressing with one dumbbell (COR with one dumbbell, flat with one dumbbell). In these trials, participants performed five repetitions with only the right arm in a continuous style. If I understand this correctly, then the style or pattern of the repetitions were not the same across trials (unilateral with 1 vs unilateral with 2). The reasoning that was provided by the authors for this difference was that athletes who train unilaterally using two dumbbells usually execute the movement in an alternating style. I can certainly appreciate the rationale for this decision, but even so, I believe this presents a potential problem when interpreting the EMG data. According to line 175, for the trials with two dumbbells, participants’ right arms were held in a fully extended position while the left arm was pressing. Which means that the right arm in these trials, despite being twice as long in duration due to the addition of the left arm, were given a period of relative rest between repetitions. In the trials with a single dumbbell, the repetitions occurred continuously, with no rest in between. My main concern is that neuromuscular fatigue may have developed differently between the two trials, which would have influenced the measures of EMG. Out of curiosity, for the trials with the two dumbbells, did the research team consider having participants unilaterally press with their right arm continuously while simply holding onto the left in an extended position to add stability? This may have made for a more controlled comparison.

My suggestion, assuming it is within the capabilities of your analysis, is to examine the EMG separately for each repetition (1 to 5) between the two trials of 1 vs 2 dumbbells. If the EMG is in changing from the first to the last repetition, and changing differently between trials, than the observed differences in EMG may be due to fatigue and not stability.

2) In the methods section, it is stated that participants were required to, “have had previous experience in participating in resistance training and weightlifting.” Can you provide any further details on how you defined “previous experience,” as it presents a wide range of possibilities in its current form?

3) This partially goes back to my first point. I realize that the trials were randomized, which would theoretically nullify the influence of neuromuscular fatigue between trials if it had occurred, but was fatigue assessed in any way during the second experimental session?

4) Was there a reason that the right arm was chosen over the participants’ dominant arm? And on that same line of thought, were there any left handed volunteers in the study? If so, did you notice any differences in their results using their non-dominant arm compared to the rest of the sample?

5) I should point out that I do not have extensive experience with ROM as an experimental measurement, so this comment may end up being trivial. With that said, in Figure 3, the standard deviations seem rather large. Are these values common with this type of movement? Were participants given any instructions regarding the depth of their repetitions? Was there any way you were able to confirm that they were truly using a full ROM?

Validity of the findings

I thought that the discussion read very well and that most of the conclusions that were brought up were both cautious and well supported by the findings of the study. My only concern goes back to my first point on the experimental design. I am not 100% convinced that some of the differences, or lack thereof, between pressing with 1 vs 2 dumbbells was due to stability and not neuromuscular fatigue.

Comments for the author

No comments

---

## Round 0.2 · Minor Revisions

· Academic Editor

Minor Revisions

Based on the reviewer comments, and in my opinion as well, there are some very minor changes to address prior to acceptance. You will notice that reviewer 1 suggests removing overly 'bold' statements, with which I agree. Please address this minor issue and resubmit.

Reviewer 1 ·

Basic reporting

N/A

Experimental design

N/A

Validity of the findings

Although I think the predictive nature of the current results, the experiment is still valid in its current form.

Comments for the author

Conclusion: Some statements in the conclusion are quite bold. Perhaps, a larger RoM is not always better for training (i.e. in the case of injury) and therefore the greater RoM would not be better.

·

Basic reporting

No Comments

Experimental design

No Comments

Validity of the findings

The revisions made to the submitted manuscript have addressed all of my previous concerns. I believe that the revised discussion is better supported by the changes made with the removal of the EMG comparisons between unilateral pressing with one or two dumbbells. I have no further comments.

---

## Round 0.3 · accepted · Accept

· Academic Editor

Accept

I am pleased to inform you that your manuscript has been accepted for publication in PeerJ.